# Targeted Radiotherapy Using Contact X-ray Brachytherapy 50 kV

**DOI:** 10.3390/cancers14051313

**Published:** 2022-03-03

**Authors:** Jean-Pierre Gerard, Arthur Sun Myint, Nicolas Barbet, Catherine Dejean, Brice Thamphya, Jocelyn Gal, Lucile Montagne, Te Vuong

**Affiliations:** 1Department of Radiation Oncology, Centre Antoine-Lacassagne, Côte d’Azur University, 06000 Nice, France; catherine.dejean@nice.unicancer.fr (C.D.); lucile.montagne@nice.unicancer.fr (L.M.); 2Clatterbrige Cancer Center, Liverpool University, Liverpool L7 8YA, UK; sun.myint@nhs.net; 3Department of Radiation Oncology, ORLAM, Bayard Lyon-Villeurbanne, 69100 Lyon, France; barbet71@wanadoo.fr; 4Department of Clinical Research-Statistics, Centre Antoine-Lacassagne, Côte d’Azur University, 06000 Nice, France; brice.thamphya@nice.unicancer.fr (B.T.); jocelyn.gal@nice.unicancer.fr (J.G.); 5Jewish General Hospital, McGill University, Montreal, QC H3T 1E2, Canada; te.vuong@mcgill.ca

**Keywords:** targeted therapy, contact X-ray, brachytherapy, organ preservation

## Abstract

**Simple Summary:**

Organ preservation is becoming a topic of great interest for rectal cancer. Radiotherapy, often in association with chemotherapy, is playing a major role in achieving tumor sterilization and long-term local control. In order to achieve this goal, a high dose (>70 Gy) of radiotherapy is necessary. To avoid radiation toxicity of such a high dose, an endocavitary approach using contact X-ray 50 kV brachytherapy (CXB) is an attractive method. Historical series and two randomized trials in Europe give good evidence of the merit of CXB. The selection of early tumors is a key prognostic factor to achieve good results. A planned organ preservation treatment using CXB followed by chemoradiotherapy can be proposed to patients with T2–T3 a–b < 3 cm in diameter. At 3 years, the chance of local control should be close to 90% with good bowel function.

**Abstract:**

Rectal adenocarcinoma is a quite radioresistant tumor. In order to achieve non-operative management (NOM) radiotherapy plays a major role. Targeted radiotherapy aiming at high precision 3D radiotherapy uses stereotactic image-guided external beam radiotherapy machines. To further safely increase the tumor dose, endocavitary brachytherapy (ECB) is an original approach. There are two different ways to perform such an ECB: contact X-ray brachytherapy (CXB) using a 50 kV X-ray generator with an X-ray tube positioned under eye guidance into the rectal cavity and high-dose-rate brachytherapy (HDRB) using iridium-192 sources positioned into the rectal cavity under image guidance. This study focused on CXB. CXB uses a small mobile generator that produces 50 kV X-rays with limited penetration. This technique is well adapted to accessible tumors of limited size and especially needs a high dose rate (≥15 Gy/minutes) for rectal tumors. It is performed on an ambulatory basis. A total dose between 80–110 Gy is delivered in 3–4 fractions over 3 to 6 weeks into a small volume (5 cm^3^). CXB was pioneered in the 1970s by Papillon using the Philips RT 50^TM^. Since 2009, the Papillon P50^TM^ has been used in 11 institutions in Europe. The OPERA Phase III trial tested the hypothesis that a CXB boost (90 Gy/3 fr) compared to an EBRT boost (9 Gy/5 fr) for T2–T3 ab < 5 cm and N0–N1 < 8 mm will increase the 3-year organ preservation (OP) rate when combined with 45 Gy/5 weeks with concomitant capecitabine. Out of more than 300 patients with tumors < 3 cm (1962–1992), Papillon reported a long-term local control close to 85%. Similar results were published in Europe and USA at that time. The Lyon R96-2 Phase III trial (2004) demonstrated that, when combined with preoperative EBRT, a CXB boost (90 Gy/3 fr) significantly increased the rate of clinical complete response (cCR) and sphincter preservation, with some patients having OP at 10 years. With more than 2000 patients treated in Europe (2010–2020) using the Papillon 50^TM^, organ preservation appears possible in close to 80% of cases in selected early T2–T3. The OPERA trial closed after 141 inclusions (2015–2020) after an independent data monitoring committee recommendation because of promising results. At the 2-year follow-up (blinded data), the rate of cCR and OP were 77% and 72%, respectively, for the 141 tumors, and for T < 3 cm (61 pts), they were 86% and 85%, respectively, with good bowel function. The final results should be available in 2022. Organ preservation using NOM appears to be a promising approach for rectal cancer. A CXB boost with chemoradiotherapy in selected early T2–T3 could become an attractive option to achieve a planned OP. This approach should be proposed to well-informed patients after discussion in an MDT.

## 1. Introduction

For more than a century, the aim of radiation therapy has always been, on one hand, to deliver 100% of the “prescribed dose” into the tumor (gross visible) volume (GTV) or into the CTV (clinical target volume) when irradiating “sub-clinical” malignant disease and, on the other hand, to deliver the lowest dose (if possible 0%) in the normal surrounding normal tissues, often specified as the “organ at risk” (OAR). The ultimate clinical aim is to control close to 100% of the tumors with less than 5% significant toxicity and good organ function preservation [1].

The main limitation to radiation dose escalation is the tolerance of the OAR. Radiation oncologists have always tried to target as much as possible of their irradiation to achieve the best therapeutic ratio. Thanks to technological improvement, this targeting has been optimized over time. The use of TeleCobalt in the 1960s was a major breakthrough to spare skin (when compared to a 200 kV generator). This was further improved with a linear accelerator and “image- and computer-guided” radiotherapy in the 1990s. It is interesting to notice that, as early as the 1920s, brachytherapy using radium tubes or needles was a great improvement in terms of targeted radiation therapy (TRT) for accessible tumors (uterus, oral cavity, skin, etc.).

In the 2020s, TRT is becoming a major field of research for rectal cancer because, until recently, radical proctectomy (RP-TME) was the only treatment to achieve a cure for this disease. However, recently, a strong research focus has been directed toward organ (rectal) preservation. In this perspective, radiotherapy plays a major role in achieving complete clinical response (cCR), sterilization of the tumor, and long-term (sustained) local control of the tumor with good preservation of the rectum and normal bowel function.

As rectal adenocarcinoma is a rather radio-resistant tumor [2], a high dose must be delivered to the tumor, but on the other hand, in order to avoid rectal toxicity, a dose over 60–65 Gy cannot be delivered through external beam radiation therapy (EBRT). This is the reason why TRT is an active field of research to improve the therapeutic ratio of irradiation. Stereotactic EBRT is a promising technological development to achieve TRT. Cyberknife^TM^ is a dedicated system that probably performs (at least at present) optimal TRT using EBRT with the help of artificial-intelligence-guided robotic technology to perform such TRT. Most of the modern linear accelerators use some form of stereotactic image-guided technique to perform high-quality TRT. This boost can often be given at the same time (same fraction) as the standard EBRT dose and is called an SIB (simultaneous integrated boost). The APHRODITE Phase II trial (ISRCTN16158514) is exploring such an SIB technique to increase the chance of NOM. In the RECTAL-BOOST trial [3], a boost dose (5 × 3 Gy) was given through a stereotactic EBRT technique, directly followed by CRT (cap 50). A total of 128 patients were included. At 2 years, there was no increase in a pathological or sustained cCR. This negative result may have been because the boost dose was too low and/or the irradiated tumors were too locally advanced (“ugly LARC”). The WW3 Phase III trial (NCT 223344) is selecting “early T1–T3” to test the hypothesis that such a boost dose (12 Gy/4 fractions) is able to increase the 2-year TME-free survival.

Another approach to safely deliver TRT is to use an endocavitary TRT (ETRT) approach. In this strategy, the radiation source is introduced (eye- or image-guided) through the anal orifice into the rectal cavity and directly facing the surface of the rectal tumor. As the radiation source is at a very short distance from the tumor, this technique is considered as brachytherapy. There are two different ways to perform such endocavitary brachytherapy: contact X-ray brachytherapy (CXB) using an X-ray generator and an X-ray tube positioned under eye guidance into the rectal cavity and high-dose-rate brachytherapy (HDRB) using iridium-192 sources that are remotely introduced in a rectal applicator positioned into the rectal cavity under image guidance. This study focused on CXB.

## 2. Materials and Methods 

Studies exploring the role of CXB were collected from the literature. First, data are presented from a historical point of view. Second, the dosimetric and technological considerations are addressed. Finally, the main clinical outcomes are given in the Results section.

### 2.1. Brief History of Contact X-ray Brachytherapy

CXB was initiated by Chaoul in Berlin in the 1930s using a Siemens generator that produced X-rays of 50 kVp. Lamarque in Montpellier (F) used it for rectal cancer in the 1950s but it was Papillon in Lyon (F) who was the pioneer of this approach using a Philips RT50^TM^ machine for selected “early” rectal cancer. With the development of linear accelerators, the manufacture of the CXB machine was discontinued. It was only in 2009 that Spanswick in the UK (Ariane Medicla System) was able to design a new 50 kVp generator, called Papillon 50^TM^, that was well adapted for rectal irradiation. Presently, 11 machines are working in Europe (UK, France, Sweden, Switzerland, Netherland, Denmark), where more than 3000 rectal cancers have been treated. Recently a new Papillon+^TM^ is available to perform rectal or skin irradiation and also breast tumor IORT [4]. Manufacture of the P+^TM^ is still in progress, with few machines available on the market. The number of radiation oncologists trained in CXB remains low. These are the two main reasons for the low uptake of this technique.

### 2.2. Basic Principles of CXB for Rectal Cancer

The rectal tumor must be accessible to the X-ray tube and the lower pole must be ≤10 cm from the anal orifice (i.e., lower-middle rectum). CXB is a fully ambulatory treatment that is accessible to all patients at any age. The treatment is performed in the knee–chest position; therefore, anteriorly and laterally located tumors are easier to target than posterior tumors, which may need a gynecologic position. Tumors involving the upper anus, if not painful, can also be treated using CXB. 

With the largest rectal applicator being 3 cm in diameter, the good candidates are rectal tumors ≤3 m ᴓ. Larger tumors can be treated after shrinkage using EBRT (or CRT) as initial treatment or sometimes using two adjacent-overlapping CXB fields.

The dose rate of the 50 kVp generator is a crucial point. As the rectal treatment is performed on an ambulatory basis in order to keep it simple and tolerable for the patient, a session must not last more than 2 min. The dose rate (prescribed and reported at the exit surface of the applicator) should be close to 15 Gy/min. To achieve such a high dose rate, the X-ray generator must be equipped with a cooling system. The Philips RT 50^TM^ uses air cooling and the Papillon system uses mineral oil cooling (Figure 1).

Depending on the machine and the applicator, the X-ray source (the anode) is located at 3 to 4 cm from the tumor or rectal mucosa. It is very important to keep this FSD (focus skin distance) at least 3 cm in order to maintain a good “penetration” of the 50 kVp beam in the tumor or tissues. This penetration mainly depends on the “inverse square law.” The consequence of this physics law is as follows: the longer the FSD, the better the penetration. This basic physics law must be mitigated with the dose rate of the machine. The optimal balance is a long FSD compatible with a high dose rate. This penetration or percentage depth dose (PDD) should be close to 50% at 5 mm (when the tumor dose is 30 Gy at the tumor surface or applicator exit, the dose at 5 mm into the tumor is 15 Gy). A long FSD (≥3 cm) is a major parameter to keep this PDD (penetration) optimal. 

CXB must be considered as ablative and adaptive irradiation. The dose per fraction (prescribed and reported at the applicator exit) is between 15 and 35 Gy, depending on the thickness of the tumor and rectal wall tolerance. On a normal mucosa, it is advised not to deliver a dose above 15–20 Gy per fraction. Such a high dose per fraction is ablative and it is possible to see a cCR (no visible tumor, no palpable tumor) 2 weeks after a dose of 30 Gy into a tumor of 2.5 cm ᴓ (Figure 2). On the other hand, when irradiating an exophytic (polypoid) tumor with CXB, as Papillon wrote in 1982 [5]: “*the tumour is destroyed layer by layer; each application treats a different layer of the tumour … The reduction in size affects both the width and the thickness of the lesion… The rapidity of the shrinkage is used as a guideline at each treatment to define the dose to be given and the interval before the next application*”. For consistency and reproducibility, the dose in the OPERA trial was 90 Gy in three fractions of 30 Gy each. It is probably better to adapt the dose at each application and to decrease the dose per fraction (and the diameter of the applicator) when the tumor is shrinking, especially when a cCR or ncCR is observed. In so doing, it was calculated by Appelt that the integral dose to the tumor is on average close to 12 Gy at each session. With a total dose of 90 Gy (D1: 35, D14: 30, D28: 25), the biologically equivalent mean dose with 2 Gy per fraction (EQD2) is close to 90 Gy (α/β tumor = 10). It is also important to stress that the dose to the rest of the rectal wall or pelvic OAR is close to zero, which explains the good tolerance of this treatment [6].

## 3. Results

The first robust data came from Papillon, who, after careful clinical selection of early polypoid tumors, often in frail elderly patients, was able to achieve a 5-year local control (or surgery-free overall survival) close to 85% in more than 300 patients with good bowel function between 1960–1990 [7]. In the years 1980–2000, similar results were published in some French and American institutions [8]. Between 1996–2001, the Lyon R 96-02 Phase III trial included 88 operable patients presenting distal rectal T2–T3 stage tumors using endo-rectal ultrasound (ERUS), who were treated with preoperative EBRT (39 Gy/13 fr/3 weeks) followed by TME surgery. Half of these treatments were randomized to receive a CXB boost (90 Gy/3 fr/4 weeks) given before EBRT. The main endpoint of the trial was sphincter-saving surgery. Out of 45 patients receiving the CXB boost, the 3-year sphincter-preserved survival was significantly improved (76% vs. 44%, *p* < 0.05). The rate of complete clinical response before surgery (defined as no palpable or visible tumor using DRE or endoscopy) was significantly improved (30% vs. 2%). Out of 45 patients treated with the CXB boost, 10 of them achieving cCR were treated with a watch-and-wait strategy with good bowel function and only one local recurrence [9]. All these data were updated after a 10-year follow-up and the difference in favor of CXB was maintained (colostomy free survival of 63% with CXB vs. 29% without CXB, *p* < 0.01) [10]. Official recognition was granted to CXB for rectal cancer T1–3 by the French authority (HAS: Haute Autorité de Santé) in 2008. The same recognition was given in the United Kingdom by NICE in 2015.

After 2002, the Philips RT 50^TM^ was not manufactured anymore. The use of CXB was discontinued in most institutions and mainly preoperative EBRT was used, often combined with chemotherapy, to reduce the risk of local relapse after TME. In the United Kingdom, Sun Myint introduced CXB in the 2000s and was able to build a dedicated Papillon Suite in 2013 for the Papillon 50^TM^, where he treated more than a hundred patients each year [11]. 

After 2010, the ICONE group (International Contact Network), including seven centers (UK: 4, France: 3), have been using CXB (Papillon 50^TM^) in three mains indications: (1) CONTEM 1 where adjuvant CXB was given after local excision for T1N0 when pejorative pathological features were observed (pT1 SM2–3, pT2, R1, budding, etc.). A good local control without surgery was observed at 3 years in 92% of 192 patients [12,13]. (2) CONTEM 2 where early T2–T3 tumors, always staged using MRI and/or ERUS, were treated with planned organ preservation and a watch-and-wait strategy if a cCR was observed after a combination of CXB + CRT (short course EBRT was used in some frail patients). At 4 years, the rate of TME-free survival was close to 85% in the United Kingdom and France [11,14,15] (Figure 3). CXB could also be used for a more palliative approach when a rectal tumor was seen in a previously irradiated pelvis or for a metastatic rectal tumor to avoid radical surgery [16]. Most of the time, CXB is well tolerated without significant toxic symptoms. One or two months after the end of treatment, the tumor disappears and may be replaced by ulceration, which can be deep (but not painful) when treating a T3 tumor with perirectal fat infiltration. Healing occurs in 3 to 4 months. Rectal stenosis is never observed. The most frequent side effect that occurs 4 to 6 months after CXB is rectal bleeding due to radiation telangiectasia. This bleeding is usually moderate (grade 1 or 2), but may sometimes require laser coagulation. It usually subsides after 1 to 3 years. Bowel function after CXB with or without CRT is usually good with a major LARS of less than 20% (Table 1). 

### Opera Phase III Trial

To test the hypothesis of the benefit of a CXB boost, the OPERA trial was initiated in 2015 and was closed in June 2020 after the inclusion of 141 patients. The inclusion criteria were operable patients, T2–T3 ab < 5 cm diameter, ≤50% circumference involvement, N0–N1 < 8 mm ᴓ, PS 0–1, distal-middle rectum (≤10 cm anal verge). The patients were stratified according to tumor diameter (< or ≥3 cm). The control treatment was neoadjuvant treatment using CRT CAP 45 (EBRT 45 Gy/25 fr/5 weeks) with concurrent capecitabine (825 mg/m^2^ BID) and an EBRT boost of 9 Gy/5 fr/1 week. The experimental treatment was the same CRT CAP 45 with the addition of a CXB boost (90 Gy/3 fr/4 weeks) given before CRT if the tumor was <3 cm or after if 3 cm or more. The endpoint of this trial was the rate of organ preservation at 3 years and the hypothesis was to improve this outcome from 20% to 40% (HR: 0.53). In July 2020, the IDMC (Independent Data Monitoring Committee) recommended to stop the accrual of patients because of encouraging data, especially in T2–3 < 3 cm, and publish the blinded data on surgical tolerability [17]. It can be seen in Table 1 (blinded data) that the TME tolerability after such neoadjuvant CXB + CRT appeared acceptable and organ preservation at 2 years was close to 85% with good bowel function in OPERA and TRESOR.

The results of OPERA at 3 years will be known in June 2022. If a benefit is observed, especially for small tumors treated with CXB first, planned organ preservation with a watch-and-wait approach when a cCR is observed could become standard and this option should be presented to the patients and discussed in a multi-disciplinary team (MDT) [18]. Total neoadjuvant treatment (TNT) is becoming popular for LARC after the OPERA, Rapido, Prodige 23, and GRECCAR 2–12 trials [19,20,21,22]. In France, before 75 years of age, Folfirinox (FIX) chemotherapy appears as an efficient (although sometimes aggressive) treatment for rectal adenocarcinoma [21,22] although so far no adjuvant or TNT has ever been shown to increase overall survival in rectal adenocarcinoma. To try to increase the chance of a planned OP in T2–T3 > 3 cm ᴓ, the TRESOR trial is under ongoing discussion in France. The reference (control) arm could be, according to GRECCAR 12, four (or six) cycles of FIX induction followed by CRT (CAP 50) and, depending on the tumor response assessment, TME in the case of partial response or W&W in the case of a cCR. Local excision could also be an option in the case of an ncCR. The experimental arm will be the same, with the addition of a CXB boost 90–110 Gy/3–4 fr/4–6 weeks delivered after FIX cycle 2 and concurrent with cycles 3–4. Adjuvant chemotherapy as in Prodige 23 is under discussion. The endpoint will be to increase the 3-year overall survival TME free from 45% to 65% and 200 patients should be included (Figure 4).

## 4. Discussion

A planned or “intended” [18,23] organ preservation in rectal cancer could be decided or proposed to a patient if, at the time of initial treatment decision, the chance of “success” is estimated close to 80% at 2 or 3 years with good bowel function and acceptable toxicity. Such an approach is a standard strategy for anal squamous cell carcinoma, which is a quite highly radiosensitive cancer (interestingly, this standard was not based on any Phase III trial vs. radical proctectomy with a permanent stoma). To achieve such an ambitious preservation goal presently in rectal adenocarcinoma (a quite highly radioresistant tumor), two parameters appear to be of importance: the possibility to safely deliver a high dose to the tumor (dose above 75 or 80 Gy) and the selection of a tumor with a not too large volume (or diameter).

To safely deliver such a high tumor dose, endorectal TRT combined with EBRT or CRT appears as an optimal strategy. Such a dose escalation using endorectal TRT could be easily tested through a Phase III trial as Lyon R96-02, OPERA, or MORPHEUS studies [24].

To accurately select and classify the candidate tumor, a very strict description of this rectal tumor is mandatory. The present TN classification is probably insufficient for an accurate and refined classification (although T2 is easier to control with irradiation than T3) and a common reproducible definition is at stake. The volume of the tumor is probably a major parameter but is difficult to measure (even with the best MRI software). An alternative could be an accurate measurement of the main diameter or height and width (in millimeters) using the circumferential extension ratio. DRE and endoscopy are key exams to perform these measurements. Imaging with MRI and ERUS is mandatory. An important benefit of imaging is to improve the measurement of the tumor penetration (thickness) in the rectal wall, peri-rectal fat, and surrounding organ (e.g., ano-rectal sphincter). Early tumors can be defined like in most guidelines [25] as T < (or ≤) 3 cm ᴓ, <1/3 circumference, and T2 or T3 a–b (penetration in the mesorectum ≤ 5 mm). The CRM (circumferential rectal margin) is a major criterion for surgery aiming at R0 resection. For radiotherapy, the CRM is less important and a T2 or T3 a–b distal and anterior close (1 mm) to the Denon Villier fascia (or even in direct contact) can be sterilized using radiotherapy as if it was the same tumor volume in the middle or posterior rectum far from the mesorectal fascia. For radiotherapy, the volume (diameter) is important but also the type of the tumor. A polypoid (exophytic) mobile tumor of 3 cm ᴓ can be sterilized with a dose close to 80 Gy. The same 3 cm ᴓ tumor with a fixed or tethered infiltrative and ulcerative type will need a dose of 100 Gy or more for the same chance of sterilization. The first one is probably well oxygenated and more radiosensitive than the ulcerative tumor, which often has hypoxic tissue components.

In contrast, an “ugly” LARC [25], i.e., T3 c–d or T4 ≥ 6 cm ᴓ or ≥2/3 rectal circumference, has a low chance of a cCR and long-term local control, even with high-dose endorectal TRT. In such a case, CRT or 5 X5 or TNT with a planned TME surgery 6–8 weeks after the end of neoadjuvant treatment is probably the best option for avoiding prolonged tumor response evaluation with little chance of a cCR and a high risk of local relapse in the case of W&W. MRI is also the best way to detect large metastatic lymph nodes (≥10 mm), which are usually not compatible with a conservative strategy.

The CEA level is also a good indicator of the tumor volume (size) and when it is elevated (maybe >10 ng, the optimal threshold is still unknown), the chance of organ preservation (for an M0 tumor) may be limited.

In many guidelines and institutions, TME first is the standard for early T2 rectal cancer [25] with no chance for organ preservation. On the other hand, TNT is recommended for LARC and, in the case of an “opportunistic” cCR, sometimes a W&W decision is taken after a long period of surveillance (4 to 6 months or more) [26]. We are approaching a time when these recommendations could be amended.

In a time of “precision medicine and tailored treatment,” the best chance to achieve organ preservation in rectal cancer is to propose such a planned strategy to selected early T1–3 or possibly to limit LARC, which is often called intermediate or “bad” LARC [25]. As radiotherapy is playing a major role to sterilize tumors (in association with medical treatment) and as endorectal TRT appears to offer the best therapeutic ratio, it is important to have a trained radiation oncologist at the time of initial MDT that is able to discuss this approach. Accurate measurement and classification of the tumor at baseline must be shared by all the specialists (gastro-enterologist, colo-rectal surgeon, radiation and medical oncologists, imaging specialists, etc.). It is important that the radiation oncologist can examine the patient and evaluate the possibility of an endocavitary approach from the start. It is also important that during the neoadjuvant treatment, especially in the case of TNT, all the specialists, not forgetting the surgeon and the “bioprobe” DRE, as it was named by Pahlman, participate in the tumor response assessment, which is a key parameter to decide between TME, local excision, or W&W. Everything can be summarized in two words: collaboration and organization between experts.

One strong argument to start treatment using CXB is that the response is assessed with DRE and rectoscopy at each CXB session by the radiation oncologist every week or at 2 weeks interval. As the dose is very high at each session (25–35 Gy), an ablative effect is expected with a rapid shrinkage of the tumor. Depending on the tumor (T1 or T2–T3 a–b), a cCR can be observed as early as day 21 or 28 after two fractions (55–65 Gy/2 fr). If an early cCR is assessed, the chance of long-term local control (after a full dose of CXB and CRT) is very high, as already mentioned by Papillon [5], and the risk of perirectal nodal failure is very low. This fast assessment of a cCR strongly reduces the medical uncertainties of the W&W approach for the oncologist, and for the patient, it relieves anxiety regarding the chance of avoiding major surgery.

If endorectal TRT is a major treatment for tumor sterilization, EBRT (with most of the time concurrent chemotherapy) is also crucial. The definition of the CTV (clinical target volume) is important to achieve a good therapeutic ratio. When organ preservation is planned, most of the time, the tumor is N0 or with limited N1 extension. It is to be remembered that the N classification, even in the best expert’s hands, is never 100% accurate. It is now well admitted [27] that the site of local pelvic relapse after TME for rectal cancer T3–4 is below the S2/S3 junction most of the time. It is also well recognized that after a W&W strategy, the majority of local relapse (or regrowth) is intramural. Therefore, it is recommended to limit the upper limit of CTV at the S2/S3 interspace when treating “early T1–T3” with planned organ intent. As the tumor never involves the whole anal canal, it is important to spare as much of this OAR as possible in order to achieve an optimal bowel function.

One of the major arguments for proposing TME radical proctectomy (and to disregard organ preservation) is to be able to control the metastatic pelvic lymphatic (or veinous) invasion. With passing time, it appears that this risk has probably been overestimated when performing W&W. More than 1000 patients were followed in the IWWD [27,28,29] after a watch-and-wait strategy and organ preservation. After more than 5 years of surveillance, 90% of the relapses (or regrowths) are located in the rectal wall at the site of the origin of the tumor, and recurrences in the perirectal fat, mesorectum, or distal pelvis appear unusual. It is reasonable, at present, to select an N0 patient (or “small N1”) for rectal preservation. It is also possible in frail, inoperable, TME-refusal patients to propose some OP strategy for a “larger N1” tumor, especially if TRT with some stereotactic EBRT can be used to selectively increase the nodal dose to 60–65 Gy (simultaneous integrated boost using MRI guided EBRT or in near-future proton therapy).

Following the Rapido trial [20], short-course radiation (with neoadjuvant chemotherapy) may be considered as a standard for T3–4 rectal cancer. This short-course regimen when used alone is probably slightly less efficient in terms of tumor sterilization than the CAPE 50 regimen but, especially in elderly, frail patients, it may have logistical advantages. Such a short course (5 X5) is fully compatible with CXB and may be used before or after CXB. In some patients, it can be proposed for easier compliance and provides a high rate of organ preservation regarding selected tumors [15].

## 5. Conclusions

Thanks to endocavitary TRT, in the coming years, organ preservation could become a standard for selected “early T1–T3” rectal tumors. Upcoming trials should test the possibility of such OP for selected intermediate LARC, depending on the tumor volume. Accurate initial screening and staging of the tumor are crucial. Close surveillance for tumor response evaluation is mandatory to assess for a cCR. Long surveillance is also mandatory to detect and salvage local recurrence (regrowth) after a W&W approach. These local recurrences should be kept as low as possible, probably below 15–20%. Organization and collaboration between colo-rectal specialists are the keys to success.

## Figures and Tables

**Figure 1 cancers-14-01313-f001:**
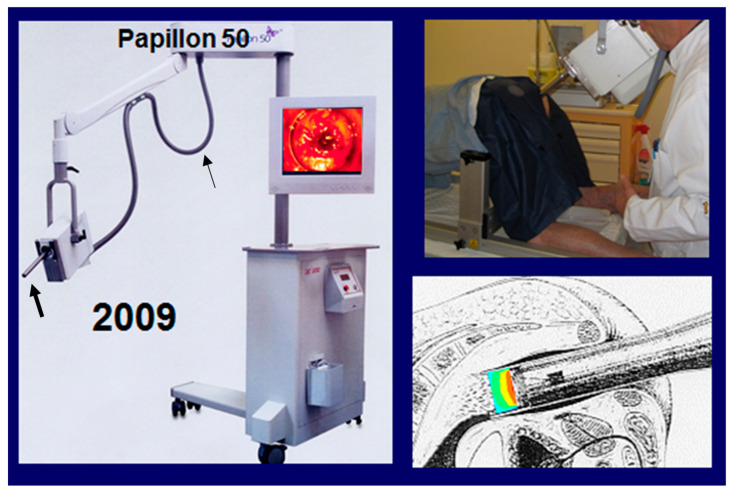
Illustration of CXB. (**Left**): The Papillon 50 TM system. Thick arrow: X-ray tube, also called a micronode. Plastic pipe (thin arrow) to circulate the mineral oil from the stand floor to the X-ray tube for anode cooling and allowing for a high dose rate (15 Gy/min). (**Upper right**): A treatment session. Patient in the knee–chest position. The rectal applicator is introduced in contact with the tumor under eye guidance and fixed (fisso arm). The X-ray tube is introduced into the rectal applicator to deliver the dose. (**Lower right**): Schematic drawing showing the X-ray tube introduced through the applicator in contact with the tumor. The FSD (focus surface distance is 4 cm). The focus is the anode (arrow) producing the 50 kV X-rays. Illustration of the dose distribution showing a sharp fall-off of the dose, which is 30 Gy at tumor surface (red) and 50% at 5 mm (green). There is nearly no dose to the rest of the rectal wall and surrounding tissues. It is a fully ambulatory treatment with good tolerance.

**Figure 2 cancers-14-01313-f002:**
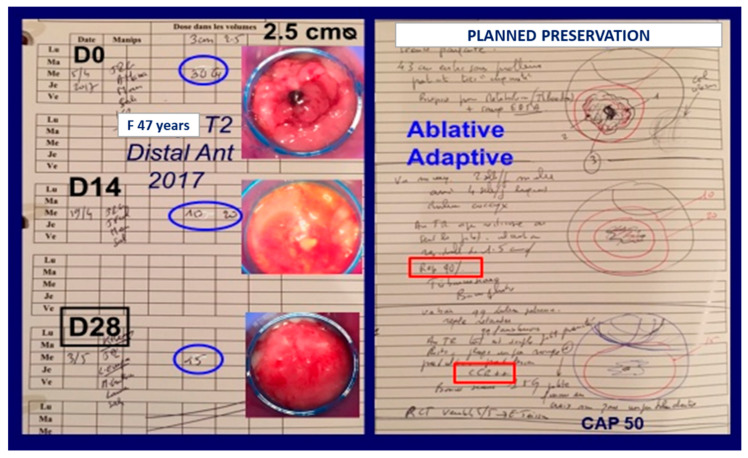
Illustration of tumor shrinkage during CXB treatment. Patient included (April 2017) in the OPERA trial in arm B1 (with CXB first). It was mainly an exophytic (polypoid) left-anterior tumor (knee–chest position: 5 to 8 o’clock). The rectoscope (blue) was 2.5 cm in diameter. Two weeks (D14) after 30 Gy (ablative dose) the tumor had shrunk by 95% and a small residual ulcer could be seen. The second fraction was given using two different applicators (adaptive strategy): a 3 cm diameter to deliver 10 Gy (in 40 s) and a second one 2.5 cm diameter delivering 20 Gy (in 1 min with a shorter FSD). Two weeks later (D28), a clinical complete response was observed. A dose of only 15 Gy (3 cm applicator) was given to the normal mucosa. One week later, chemoradiotherapy was initiated (45 Gy/5 weeks with concurrent capecitabine). Four years later in 2021, the patient is alive and well with good bowel function.

**Figure 3 cancers-14-01313-f003:**
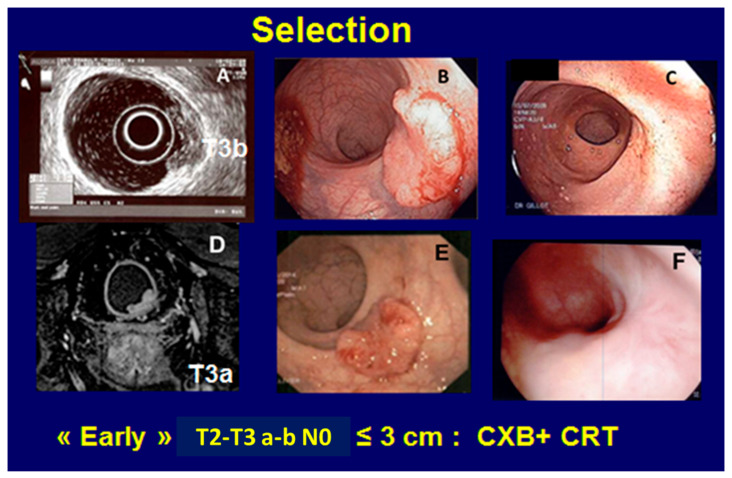
Example of T3 rectal adenocarcinoma suitable for organ preservation and treated using CXB 50. The tumor was 3 cm or less, <1/3 rectal circumference (**A**–**D**), partly exophytic with moderate central ulceration (**B**–**E**) and could easily be incorporated into the rectal applicator of 3 cm diameter (low-middle rectum), especially if located in the anterior or lateral part of the rectum (when the patient is in the knee–chest position). Normal rectal wall one year after treatment (**C**–**F**).

**Figure 4 cancers-14-01313-f004:**
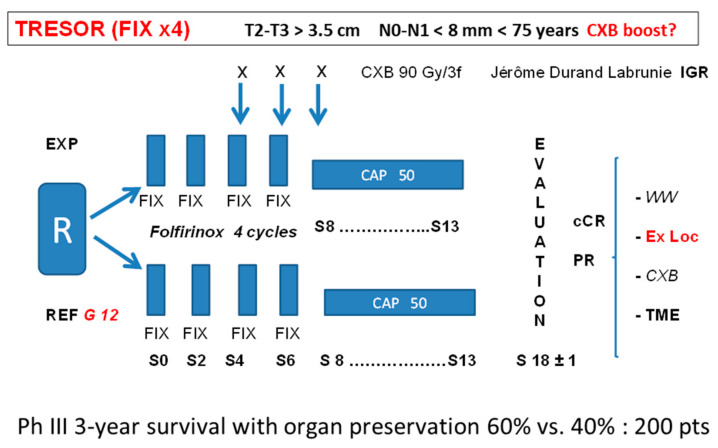
Tentative TRESOR trial.

**Table 1 cancers-14-01313-t001:** OPERA trial. Blinded data with a median follow-up of 2 years: data analyzed in April 2021. 142 patients in this analysis with 61 patients presenting a tumor < 3 cm diameter. Ant Res: anterior resection; APR: abdomino-perineal resection; cCR: complete clinical response; LARS: LARS score for rectal function.

Variable	Total (141)	T < 3 cm ᴓ (61)
Age (median) year	68	68
T2/T3 ab	91/50	53/8
Distal/middle rectum	105/36	47/14
Ant Res./APR	21/9 (30)	4/2 (6)
ypT0-is	4	0
ypN1	3	0
R0	28	6
Hospital stay (days)	9.5	
Second surgery	3	
Medical toxicity	4	
Death 30 days	0	
cCR (W18–24)	77%	86%
Organ preservation (2 years)	72%	85%
LARS < 3	85%	87%

## Data Availability

All data of the study are available upon request (for confidentiality reasons) at CENTRE ANTOINE LACASSAGNE, Nice, France.

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
