# Peer review of "Targeted Radiotherapy Using Contact X-ray Brachytherapy 50 kV"

_cancers, 2022, doi:10.3390/cancers14051313_

Round 1

Reviewer 1 Report

This is a nice review article outlining the current status of the role of different delivery methods for radiation boost and how it relates to organ preservation in the era of TNT. Minor english language and grammar checks would make it a little easier to read but overall it is fine.

As it is a review article, materials and methods should include how the included studies were identified rather that a historical description. This section should be modified. 

Lastly , the topic of use of short course radiation in TNT (which does not utilized concurrent cap and long course radiation with a boost should be acknowledged and further addressed in greater detail in the discussion. 

Overall, this is a nice paper and will generate much interest to readers.

The conclusions are consistent with the evidence provided and do address the main question posed.

The references are appropriate.

Author Response

Dear Isidora

I have revise my manuscript « Targeted radiotherapy using Contact x-ray brachytherapy 50 kV» according to the referees’ comments.

I was not able to find and use the « Track Changes » function. All the modifictions are in red and easily visible. A final version in black is also available.

Thank you for the reviewers comments.

Find below my point by point responses.

Reviewer 1 :

I have made some english language  improvements but feel free to do more if deemed appropriate.

I have modified the section material-Methods  to fit with the presentation of a review article.

I have added a special paragraph dealing with short course irradiation.

Some figures in the text and in Tables have been modified because we have been updating recently the data of the  OPERA trial.

Reviewer 2 Report

This is a review article on the use of X-ray brachytherapy in rectal cancer.

CXB is a dose-escalation and OAR-sparing RT technique that plays an important role in rectal cancer when the goal of treatment is organ preservation.

The article reports the rationale, history of the techniques, basic principles, tumour criteria for treatment and results of studies that have adopted this technique.

There are no major comments, but some more information could be added:

the exclusion criteria for CXB should be described (tumour involving the anal canal? Differences between tumour in the anterior and posterior rectal wall? others?);

data on patient compliance, long-term side effects and functional outcome measurements should also be reported: can all patients be treated with this technique? Type and frequency of late side effects such as bleeding, stenosis, pain. What is the higher rate of LARS compared to standard TME? (there is some evidence that 30% of patients treated with local excision still experience major LARS)

Although very promising and apparently low cost and low burden, this technique is only adopted by a few centres, can the Authors comment on the reasons for the low uptake of this technique?

There are several other small typographical inaccuracies in the text that should be corrected (capital letters, spaces, forgotten brackets...)

For example:

Page 3 line 139 is written Medicla

Page 5 line 207 says ulta

Page 6 line 262 says 'in From OPERA to TRESOR' (I do not understand)

Table 1 in the legend mentions Ov Surv, however the table does not show survival data; data on hospital stay, medical toxicology and death at 30 days are missing for the 6 patients operated on with an initial tumour diameter < 3 cm

Figure 1: some words are in French

Figure 2: no illustration of dose distribution, although described in the legend

Figure 3: in the legend line 2 there is the word 'figure' between CXB and 50, is this a typing error?

References should be standardized

Author Response

Dear Isidora

I have revise my manuscript « Targeted radiotherapy using Contact x-ray brachytherapy 50 kV» according to the referees’ comments.

I was not able to find and use the « Track Changes » function. All the modifictions are in red and easily visible. A final version in black is also available.

Thank you for the reviewers comments.

Find below my point by point responses.

Reviewer 2 :

Precison was given regarding the use of CXB for tumor involving the anal canal and anterior or posterior rectum. It was mentionned  that CXB can be used in all patients at any age.

Toxicity and side effect were described stressing the risk of rectal bleeding and some ulceration. No rectal stenosis.

Bowel function and LARS score are described .

Slow manufacturing of the CXB machine and few trained radiation oncolgists was mentionned as the reason for low uptake

Mical and ulta were corrected

OPERA –TRESOR was deleted

Table 1 :  Ov Surv was deleted. We don’t have separate data for T< 3 cm.

Fig 2 : it is explained in the legend that the 100 % dose is the red colour and the 50% is the green. This illustrate the rapid fall of of the dose within 5 mm.

Fig 3   :  Figure was deleted

Reference were standardized using End Notes.